# Virus infection is controlled by hematopoietic and stromal cell sensing of murine cytomegalovirus through STING

**Sytse J Piersma[1]\*, Jennifer Poursine-Laurent[1], Liping Yang[1], Glen N Barber[2], Bijal A Parikh[3], Wayne M Yokoyama[1]**

[1]Division of Rheumatology, Department of Medicine, Washington University School of Medicine, St. Louis, United States; [2]Department of Cell Biology and Sylvester Comprehensive Cancer Center, University of Miami School of Medicine, Miami, United States; [3]Department of Pathology and Immunology, Washington University School of Medicine, St. Louis, United States

**Abstract** Recognition of DNA viruses, such as cytomegaloviruses (CMVs), through pattern-recognition receptor (PRR) pathways involving MyD88 or STING constitute a first-line defense against infections mainly through production of type I interferon (IFN-I). However, the role of these pathways in different tissues is incompletely understood, an issue particularly relevant to the CMVs which have broad tissue tropisms. Herein, we contrasted anti-viral effects of MyD88 versus STING in distinct cell types that are infected with murine CMV (MCMV). Bone marrow chimeras revealed STING-mediated MCMV control in hematological cells, similar to MyD88. However, unlike MyD88, STING also contributed to viral control in non-hematological, stromal cells. Infected splenic stromal cells produced IFN-I in a cGAS-STING-dependent and MyD88-independent manner, while we confirmed plasmacytoid dendritic cell IFN-I had inverse requirements. MCMV-induced natural killer cytotoxicity was dependent on MyD88 and STING. Thus, MyD88 and STING contribute to MCMV control in distinct cell types that initiate downstream immune responses.

\*For correspondence:
spiersma@wustl.edu

**Competing interests:** The authors declare that no competing interests exist.

## Introduction

Viral infections can be detected by specialized pattern recognition receptors, which recognize viral structures that are unique or otherwise absent in the subcellular location where they are detected. Nucleic acids from DNA-viruses can be detected in various organelles during infection. Some DNA viruses pass through endolysosomes where viral DNA can be recognized by toll-like receptors (TLRs), in particular TLR9, which signals through MyD88 and induces a type I interferon (IFN-I) response (*Hemmi et al., 2000*; *Motwani et al., 2019*). In the cytosol, infection results in exposure of viral DNA that can be recognized by cytosolic DNA sensors including cyclic GMP-AMP synthase (cGAS) and absent in melanoma 2 (AIM2) inflammasome (*Bürckstümmer et al., 2009*; *Schoggins et al., 2014*; *Sun et al., 2013*; *Tan et al., 2018*). cGAS signals through STING and initiates an IFN-I response, whereas AIM2 activates caspase I and instigates an IL-1β and IL-18 responses. The TLRs and AIM2 pathways are primarily active in specific immune cell types. In contrast, the STING-cGAS pathway appear to be active in a broader range but not all cell types (*Motwani et al., 2019*; *Thomsen et al., 2016*). Yet, it is unclear how activation of these pathways in different cell types contributes to viral control.

The IFN-I that is produced in response to viral recognition plays a central role in protection against acute infection. IFN-I mediates its anti-viral effects through stimulation of the interferon receptor, comprising of IFNAR1 and IFNAR2, and downstream STAT molecules. The resulting IFN-stimulated genes (ISGs) induce an anti-viral state, affecting cell survival and viral replication

(*González-Navajas et al., 2012*; *McNab et al., 2015*). In addition, IFN-I is critical for orchestrating the subsequent innate and adaptive immune responses, through modulation of cell attraction, activation, and priming. Although human deficiencies in the IFN-I pathway are very rare, evidence suggest that IFN-I could protect against viral infections in humans. Individuals with mutations in *IFNAR2* and *STAT2* have relatively mild symptoms after infection, even though they can develop severe illness in response to live vaccines and can have recurrent viral infections (*Duncan et al., 2015*; *Hambleton et al., 2013*; *Moens et al., 2017*). However, these deficiencies likely do not completely nullify IFN-I effects because IFNβ can signal through IFNAR1 without requiring IFNAR2 and IFN-I can signal through STAT2-independent pathways (*de Weerd et al., 2013*; *González-Navajas et al., 2012*). In addition, other loss-of-function mutations that affect the IFN-I pathway have been described to enhance susceptibility to virus infection, including IRF7, IRF3, IRF9, and STAT1 (*Andersen et al., 2015*; *Bravo García-Morato et al., 2019*; *Chapgier et al., 2009*; *Ciancanelli et al., 2015*; *Hernandez et al., 2018*; *Kong et al., 2010*; *Thomsen et al., 2019a*; *Thomsen et al., 2019b*). Thus, IFN-I is critical to control viral infections, but it remains unclear what pathways contribute to viral control.

In this regard, studies of infections with the beta-herpesvirus cytomegalovirus (CMV), have been informative. Infection with human CMV (HCMV) is nearly ubiquitous worldwide (*Cannon et al., 2010*). HCMV is controlled and establishes latency in healthy individuals, but HCMV can cause life-threatening disease in immunocompromised patients (*Griffiths et al., 2015*). Despite a broad tropism that allows CMV to infect a wide range of cell types, CMV is highly species-specific (*Krmpotic et al., 2003*; *Sinzger et al., 2008*). Murine CMV (MCMV) in particular shares key features with HCMV and has been instructive for dissecting cytomegalovirus pathogenesis (*Krmpotic et al., 2003*; *Picarda and Benedict, 2018*). Indeed, a recent case study described a patient with deficiencies in both *IFNAR1* and *IFNGR2* who presented with bacteremia and CMV viremia (*Hoyos-Bachiloglu et al., 2017*). Consistent with these findings, mice deficient in *Ifnar1* and *Ifngr1* are highly susceptible to MCMV in 129Sv and C57BL/6 strains (*Gil et al., 2001*; *Presti et al., 1998*). *Ifnar1* deficiency in isolation resulted in a 100-fold increased MCMV susceptibility whereas *Ifngr1* deficiency did not, indicating that IFN-I plays a dominant role in controlling acute CMV infections. IFN-I production during acute MCMV infection is biphasic; initial IFN-I production peaks at 8 hr post infection (p.i.) with a second peak at 36–48 hours p.i. (*Delale et al., 2005*; *Schneider et al., 2008*). STING has been implicated in the initial IFN-I response. STING-deficient mice have decreased systemic IFNβ at 12 hours p.i. and 5-fold increased viral load at 36 hours p.i. (*Lio et al., 2016*). A recent study implicated Kupffer cells to be the main source for IFNβ in the liver 4 hours p.i. (*Tegtmeyer et al., 2019*). Besides the aforementioned immune cells, stromal cells are thought to be a major source for IFN-I in the spleen at 8 hours p.i. (*Schneider et al., 2008*). By contrast, MyD88-dependent pathways have been implicated in IFN-I production during the second wave (*Delale et al., 2005*; *Krug et al., 2004*). IFN-I production by plasmacytoid dendritic cells (pDCs) is dependent on TLR7 and TLR9 (*Hokeness-Antonelli et al., 2007*; *Krug et al., 2004*; *Zucchini et al., 2008b*). Consistent with the role of pDCs in IFN-I production, MyD88 is required in the hematological compartment in bone marrow chimeras (*Puttur et al., 2016*). However, it has been unknown which sensing pathway is responsible for IFN-I induction in the stroma and how each contributes to control MCMV infection in different tissues.

Besides its direct anti-viral effects, IFN-I is crucial for optimal NK cell function during viral infection (*Orange and Biron, 1996*). NK cells play a critical role in controlling MCMV infection in C57BL/6 mice, which is dependent on interactions between the Ly49H NK cell activation receptor and its MCMV-encoded ligand m157 (*Arase et al., 2002*; *Brown et al., 2001*; *Smith et al., 2002*). However, this interaction is not sufficient to allow NK cell control of MCMV infection. IL-12 and IFN-I produced early during MCMV infection induce granzyme B and perforin protein expression in NK cells (*Fehniger et al., 2007*; *Nguyen et al., 2002*; *Parikh et al., 2015*), which allows them to efficiently kill virus-infected cells upon recognition of m157 through Ly49H (*Parikh et al., 2015*). IL-12 and IFN-I also induce IFNγ transcription, which is required for activation receptor-dependent IFNγ production (*Piersma et al., 2019*). In the absence of MyD88, Ly49H+ NK cells can compensate for suboptimal IFN-I production (*Cocita et al., 2015*), suggesting that low levels of IFN-I can still enhance NK-mediated control of MCMV. However, which MCMV-sensing pathway contributes to the NK cell response is still unclear.

In the current study, we analyzed survival, viral titers, IFN-I production and NK cell responses in mice deficient in MyD88, STING or both. We also determined the contribution of both signaling pathways in different tissues to their anti-viral effects, and elucidated a role for cGAS in these responses.

## Results

### MyD88 and STING-dependent pathways control MCMV infection in vivo

We set out to investigate the relative contribution of STING- versus MyD88-dependent pathways in controlling MCMV infection by analyzing the morbidity and mortality in wildtype C57BL/6 (WT), MyD88-deficient (MyD88 KO), and STING-deficient (STING GT) mice as well as mice deficient in both MyD88 and STING (DKO) that were infected with 50,000 PFU MCMV (*Figure 1*). Consistent with previously published data (*Lio et al., 2016*), WT mice lost approximately 10% of weight by 3 days p.i. after which they recovered (*Figure 1A*). Here we observed that STING GT mice showed more pronounced weight loss compared to WT mice, but were also able to recover. Consistent with previously published data (*Delale et al., 2005*), MyD88 KO mice showed delayed weight loss as compared to WT mice, indicating that the initial weight loss in WT mice was caused by immunopathology mediated by MyD88. The weight curves of DKO mice overlapped with MyD88 KO mice, suggesting that STING-mediated responses do not contribute to immunopathology. Both WT and STING GT mice were able to control and survive viral infection upon challenge with MCMV (*Figure 1B*). MyD88 KO mice were moderately resistant to the infection as 37% of the mice died between days 6 and 7. In contrast, the majority (70%) of DKO mice succumbed to the infection. Thus, both STING and MyD88 significantly contribute to control of MCMV infection in vivo.

### STING contributes in both the hematological and radio-resistant compartments in controlling viral load

To investigate the contribution of STING and MyD88 in different organs, we analyzed viral loads in the spleen and liver, the initial organs of replication after infection (*Hsu et al., 2009*; *Sacher et al., 2008*). In the spleen, we observed a modest but significant increase (6.9-fold) in viral load in MyD88 KO mice two days p.i., whereas the spleens of DKO mice contained 84-fold higher viral copies compared to WT controls (*Figure 2A*). Consistent with previous studies (*Lio et al., 2016*), we observed a 3.5-fold increase in viral load in the spleens of STING GT versus WT controls, but this difference did not reach statistical significance. By 5 days p.i. we observed an 85-fold increase in viral load in the

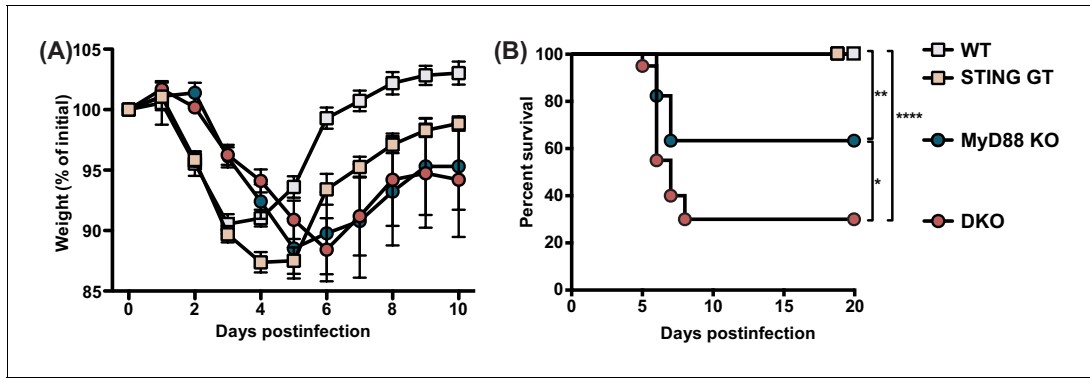

**Figure 1.** MyD88 and STING control morbidity and mortality during MCMV infection. Mice were infected with 50,000 PFU MCMV WT-1, weight loss and survival was monitored over time. (A) Weight loss over time in wildtype (n = 12), STING-deficient (STING GT, n = 21), MyD88-deficient (MyD88 KO, n = 9) and mice deficient in both STING and MyD88 (DKO; n = 14). The numbers indicate the number of mice at the start of the experiment, weight loss of surviving mice at each timepoint is plotted. (B) Survival curves of wildtype (n = 17), STING GT (n = 18), MyD88 KO (n = 17) and DKO mice (n = 20). Cumulative data of 3 independent experiments. Error bars indicate SEM; *p<0.05, **p<0.01, ****p<0.0001.

The online version of this article includes the following source data for figure 1:

**Source data 1.** MyD88 and STING control morbidity and mortality during MCMV infection.

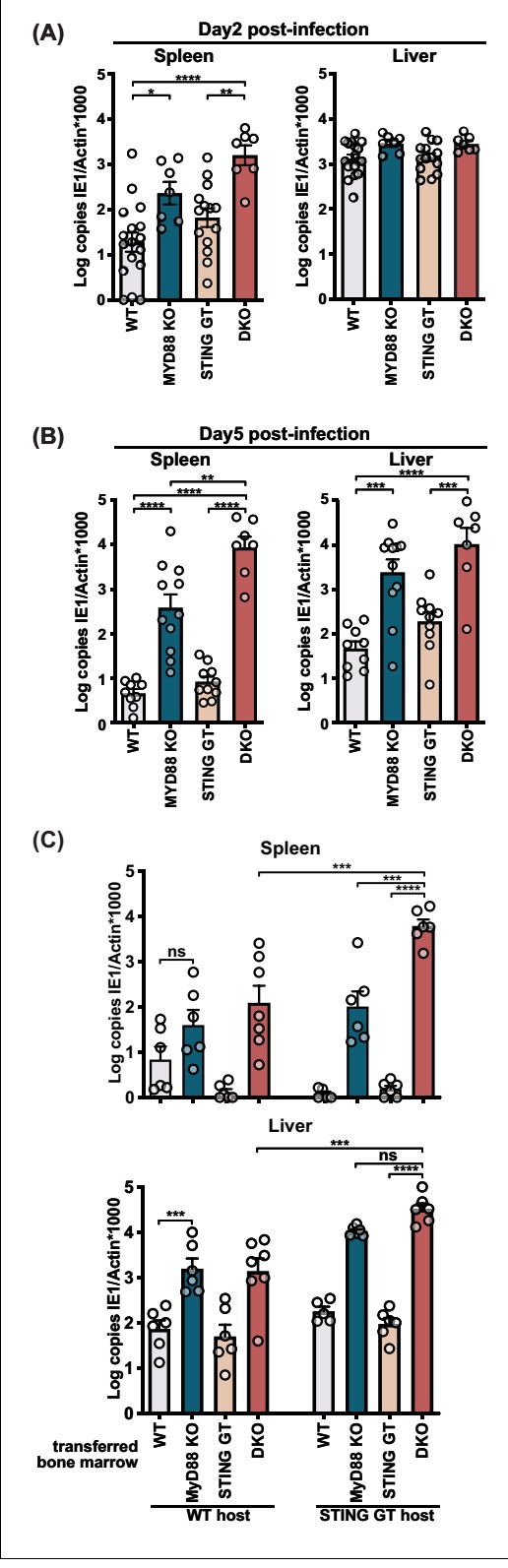

**Figure 2.** STING contributes to control of MCMV in the hematological and stromal compartment, whereas MyD88 in the hematological compartment potently controls infection. Mice were infected with 50,000 PFU

*Figure 2 continued on next page*

spleens of MyD88-deficient and 1901-fold increase in viral load in DKO, both as compared to WT controls (*Figure 2B*). We did not observe significant differences in STING-deficient animals, but we observed a 23-fold increase in viral load in DKO spleens compared to MyD88 KO, indicating that STING contributes to viral control in the absence of MyD88. In the liver, we were unable to detect significant differences in viral load at 2 days p.i. (*Figure 2A*). By day 5, we observed a 221-fold increase in DKO and 51-fold increase in MyD88 KO viral load compared to WT controls (*Figure 2B*). Taken together, these data indicate that the STING and MyD88 pathways contribute to viral control at early timepoints, particularly in the spleen and to a lesser extent in the liver.

MyD88 has been reported to be required in the hematological compartment, but not in the radio-resistant compartment (*Puttur et al., 2016*), yet it is unclear which compartment(s) requires STING-dependent pathways. To investigate the contributions of STING and MyD88 dependent pathways in these compartments, we generated bone marrow (BM) chimeras of either or both knockout BM into irradiated WT or STING GT hosts and analyzed viral load day 5 p.i. (*Figure 2C*). While WT hosts reconstituted with WT BM controlled viral load similar to WT control mice (*Figure 2C* vs *Figure 2—figure supplement 1A*), reconstitution of WT hosts with MyD88-deficient BM resulted in elevated viral loads compared to reconstitution with WT BM (*Figure 2C*), consistent with previously published results (*Puttur et al., 2016*). We also observed that the contribution of MyD88 in the hematological compartment was particularly overt in the absence of STING, revealed by comparison of DKO BM into STING GT host versus STING GT BM into STING GT host, which resulted in a 3882-fold difference in the spleen and 344-fold in the liver, respectively. STING also had antiviral effects in the hematological compartment, evident by comparing DKO BM into STING GT host versus MyD88 KO BM into STING GT host, which revealed a 59-fold difference in the spleen. STING played also a role in the radioresistant compartment in both spleen and liver, revealed by comparison of DKO BM into WT host versus DKO BM into STING GT host, for which we observed a 49-fold and 24-fold differences in the spleen and liver, respectively. Jointly, the BM chimeras reveal an evident role for MyD88 in the hematological compartment, while STING contributes to viral control in both

*Figure 2 continued*

(A) and (B) or 20,000 PFU (C) MCMV. Viral load was quantified 2 days (A) or 5 days (B) and (C) p.i. (C) Indicated bone marrow was adoptively transferred into irradiated wildtype (WT) or STING-deficient (STING GT) hosts. Bone marrow chimeras were infected 6 weeks post transfer and viral load was analyzed 5 days p.i. Each panel shows cumulative data of 2 independent experiments. Error bars indicate SEM; ns, not significant, *p<0.05, **p<0.01, ***p<0.001, ****p<0.0001.

The online version of this article includes the following source data and figure supplement(s) for figure 2:

**Source data 1.** STING contributes to control of MCMV in the hematological and stromal compartment, whereas MyD88 in the hematological compartment potently controls infection.

**Figure supplement 1.** Viral load for 20,000 PFU infection 5 days p.i. and extended statistical analysis for bone marrow chimeras.

the hematological and radio-resistant compartments, most explicitly in the spleen.

## Multiple cell populations produce IFN-I in response upon MCMV infection

Type I IFNs are induced in response to triggering of pathogen recognition receptors that signal through MyD88 and STING and are key players in the initial anti-viral response. To investigate the IFN-I response in virus-infected cells we made use of a reporter virus that expresses GFP under the IE1 promoter (*Henry et al., 2000*). We analyzed initial times (8- and 36 hours p.i.) and focused on stromal cell and CD11c+ dendritic cell (DC) populations, which are the major cell types infected at these timepoints (*Hsu et al., 2009*). Consistent with previous published data, we detected infection of the stromal cell but not CD45+CD11c+ compartment at 8 hours p.i. (*Figure 3A*). At 36 hours p.i., the percentage of infected stromal cells increased sub-

stantially and infected CD45+CD11c+ cells were detected as well. Infected CD45+CD11c+ cells included among others conventional dendritic cells (cDC) and plasmacytoid DC (pDC) populations. Based on these data, we sorted and analyzed infected and uninfected populations at 36 hours p.i. for IFNα and IFNβ transcripts by quantitative PCR (*Figure 3—figure supplement 1*). The infected stromal cells (GFP+) specifically expressed *Ifna* and *Ifnb1* transcripts, which were not detected in the uninfected (GFP-) cells (*Figure 3B*). Infected CD11c+ cells also expressed transcripts for *Ifna* and *Ifnb1* but high levels of *Ifna* transcripts were also detected in GFP- CD11c+ cells isolated from MCMV-infected animals, while *Ifnb1* transcripts were much lower in this population compared to infected CD11c+ cells (*Figure 3C*). Thus, *Ifnb1* expression correlated with infection status in CD45-

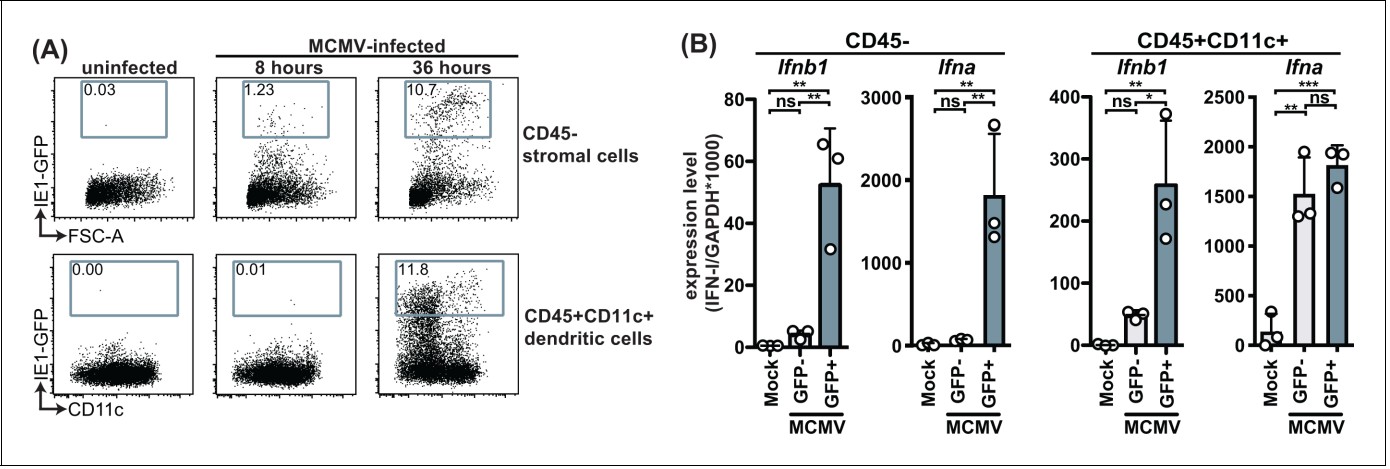

**Figure 3.** MCMV-infected cells specifically produce IFNβ upon infection. WT mice were infected with 100,000 PFU MCMV IE1-GFP reporter virus. (A) Analysis of GFP expression in CD45- stromal cells and CD45+CD11c+ DC at 8 hr and 36 hours p.i. (B) GFP+ and GFP- stromal cells and DC were FACS-sorted 36 hours p.i. and *Ifnb1* and pan-*Ifna* transcript levels were quantified by real-time PCR. Both panels show representative experiments from two independent experiments. Error bars indicate SD; ns, not significant, *p<0.05, **p<0.01, ***p<0.001.

The online version of this article includes the following source data and figure supplement(s) for figure 3:

**Source data 1.** MCMV-infected cells specifically produce IFNβ upon infection.

**Figure supplement 1.** Gating strategy and purity of sorted cell populations.

and CD45+CD11c+ cells, while *Ifna* did not correlate with infection in CD45+CD11c+ cells. Based on these data we chose to investigate the role of STING and MyD88 on IFNβ production by different cell types.

## IFNβ is produced by pDCs in a MyD88-dependent but STING-independent manner during infection

To evaluate the role of STING and MyD88 on IFNβ production by individual cells upon infection, we backcrossed MyD88 KO, STING GT, and DKO mice to the IFNβ-YFP reporter mouse (*Scheu et al., 2008*). Approximately 20% of the pDCs were YFP+, indicating at least this percentage of pDCs produced IFNβ in response to MCMV infection, whereas much fewer cDCs produced IFNβ because less than 1% of cDCs were YFP+ (*Figure 4B*). Consistent with previous studies of primary pDC in vitro and in vivo (*Krug et al., 2004*; *Tomasello et al., 2018*; *Zucchini et al., 2008b*), we observed that the production of IFNβ by pDCs was solely dependent on MyD88 because MyD88 KO mice were unable to induce detectable YFP (IFNβ) in pDCs. By contrast, here we found that STING GT mice did not significantly affect pDC IFNβ production, indicating that MyD88 functions in these cells without requiring STING-dependent pathways. On the other hand, both STING and MyD88 seemed to affect IFNβ reporter levels in the few YFP+ cDCs, although the differences were not significant (*Figure 4B*). Nonetheless, these results indicate that MyD88-dependent sensing of MCMV dictated the IFNβ response in pDCs, but it remained unclear how MyD88- and STING-dependent pathways contribute to IFNβ production in stromal cells.

## IFNβ is produced by stromal cells in a STING-dependent but MyD88-independent manner during infection

Since we were unable to find YFP+ infected stromal cells, which might be due to a detection level issue in these cells in vivo (*Figure 4*), we turned to in vitro infection of primary splenic fibroblasts and challenged them with MCMV at MOI 5 (*Figure 5A*). Indeed, splenic fibroblasts readily expressed 8000-fold increase in *Ifnb1* transcripts by qPCR at 8 hours p.i. To determine the role of key innate sensing components, we used mouse embryonic fibroblasts (MEFs) that were genetically deficient in these components. Consistent with primary splenic fibroblasts, MEF expressed *Ifnb1* transcripts upon MCMV infection (*Figure 5B*), reaching levels similar to those detected in primary splenic fibroblasts. We further observed that *Ifnb1* expression was independent of MyD88 and TRIF, indicating that TLRs do not contribute to IFNβ production in fibroblasts even though *Ifnb1* expression was dependent on IRF3/7 and TBK1, which is consistent with cytosolic sensing of MCMV. Using MEF lines with two different mutations in STING (*Ishikawa and Barber, 2008*; *Sauer et al., 2011*), we found that the IFNB1 response was instead dependent on STING. However, IFNB1 production was independent of MAVS (also known as CARDIF and IPS-1), suggesting that the IFN-response is independent of the cytosolic RNA sensors (*Tan et al., 2018*). Finally, we investigated the role of cytosolic DNA sensors and found that fibroblast sensing of MCMV was dependent on cGAS, but independent of ZBP1 and DNA-PK. To confirm that the cGAS pathway is involved in adult splenic stromal cells, we analyzed *Ifnb1* expression in cGAS-deficient primary splenic fibroblasts (*Figure 5C*). Indeed, cGAS-deficient splenic fibroblasts were unable to express *Ifnb1* in response to MCMV challenge, indicating that the STING-cGAS-dependent pathway is responsible for the IFNβ response in the stromal cell compartment. To validate that these pathways are also involved in IFNβ protein production and secretion, we analyzed cell culture supernatants at 48 hours p.i. with MCMV MOI 0.5 (*Figure 5D*). WT MEF secreted IFNβ in response to MCMV infection, but neither STING nor cGAS-deficient MEFs produced IFNβ. Collectively, these results strongly suggest that the stromal cell compartment produces IFNβ in a STING-cGAS dependent but MyD88-independent manner.

## MyD88 and STING contribute to NK cell cytolytic potential

We previously reported that both IFN-I and IL-12 act directly on NK cells to induce perforin (Prf) and granzyme B (GzB) protein levels, thereby increasing NK cell cytolytic potential, which was required for Ly49H-dependent control of MCMV infection (*Parikh et al., 2015*). Moreover, IL-12 production in response to MCMV has been reported to be dependent on MyD88 (*Krug et al., 2004*), and thus contributed to the phenotypes observed in MyD88 KO mice independent of IFN-I. Here we investigated the role of MyD88 and STING in increasing NK cell reactivity during MCMV infection.

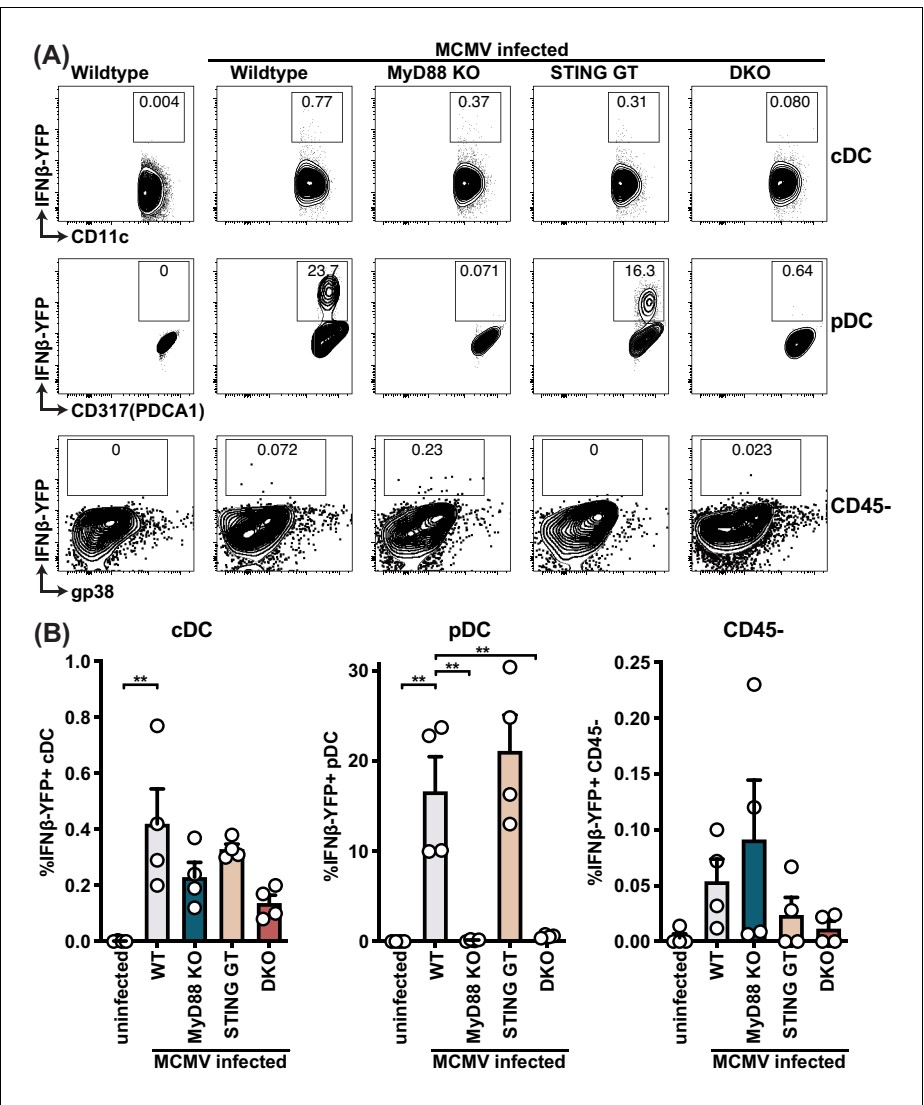

**Figure 4.** pDCs produce IFNβ in a MyD88-dependent but STING-independent manner in IFNβ-YFP reporter mice. IFNβ-YFP reporter mice were backcrossed to MyD88- (MyD88 KO), STING- (STING GT) and double-deficient (DKO) mice. Animals were infected with 200,000 PFU WT1 MCMV and analyzed 48 hr post infection. Spleens were digested to a single cell suspension, stained and analyzed by flow cytometry. Error bars indicate SD; **p<0.01. The online version of this article includes the following source data and figure supplement(s) for figure 4:

**Source data 1.** pDCs produce IFNβ in a MyD88-dependent but STING-independent manner in IFNβ-YFP reporter mice.

**Figure supplement 1.** Gating strategy for analysis of IFNβ-YFP reporter mice.

Consistent with previous reports (*Fehniger et al., 2007*; *Orange et al., 1995*; *Parikh et al., 2015*), we observed increased levels of NK cell GzB, Prf and IFNγ at 48 hours p.i. (*Figure 6A*). At this time point, NK cell production of IFNγ is reportedly dependent on IL-18, which signals through MyD88 (*Adachi et al., 1998*; *Pien et al., 2000*). Indeed, NK cell IFNγ production was dependent on MyD88, whereas STING did not impair IFNγ production, and rather increased the IFNγ response (*Figure 6B*). This potentially could be due to a relatively small increase in viral load at these timepoints. Expression of both GzB and Prf followed a similar pattern, as the vast majority of increased expression was dependent on MyD88, whereas STING did not overtly contribute to the production of these lytic proteins (*Figure 6B*). Finally, we analyzed NK cell cytolytic capacity using a 3 hr in vivo target cell rejection assay. We previously reported that MCMV increased m157-target cell rejection in an IL-12- and IFN-I-dependent manner (*Parikh et al., 2015*). Consistent with our previous data, m157-specific

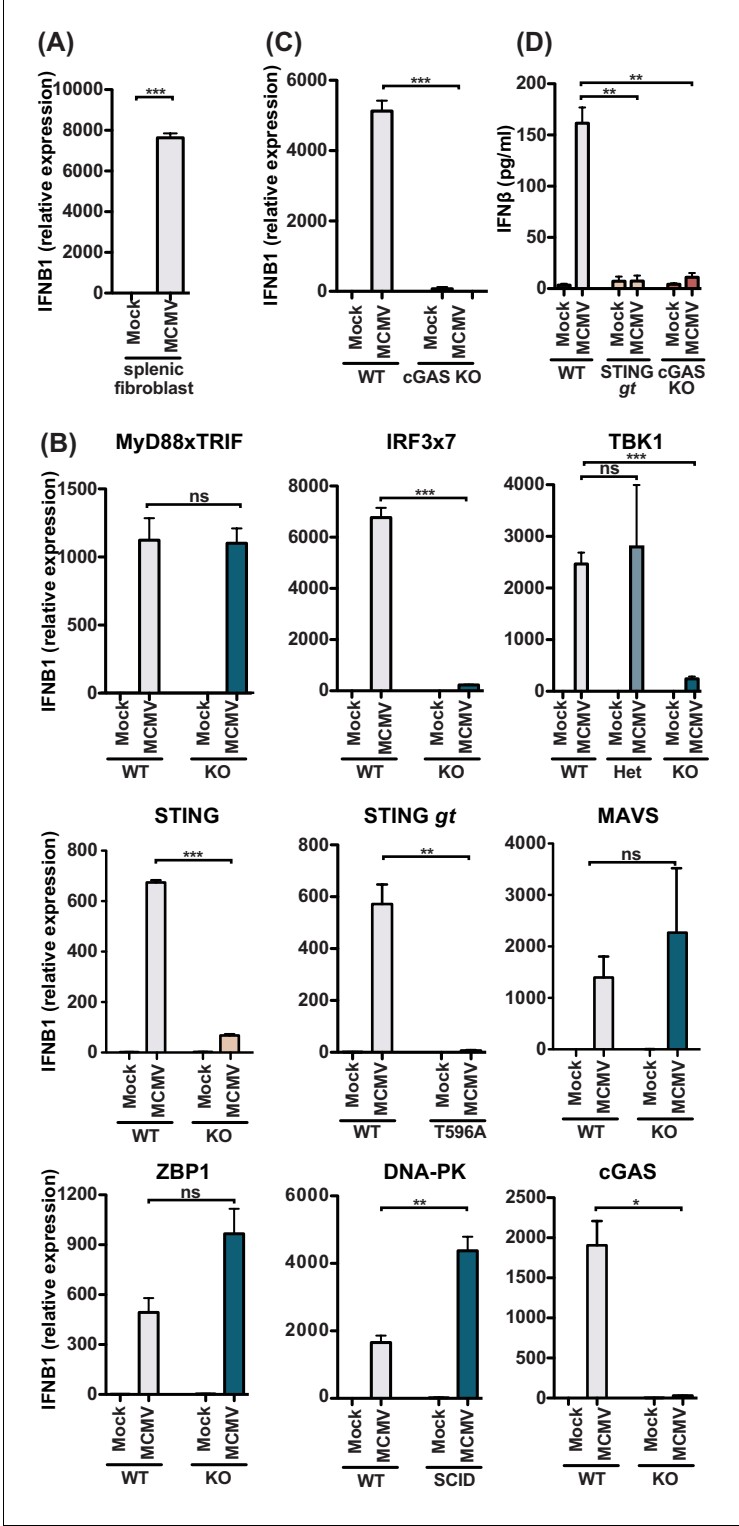

**Figure 5.** MCMV-induced fibroblast IFNβ is triggered by cGAS-STING-dependent but MyD88-Trif-MAVS-independent mechanisms. (**A**) IFNB1 mRNA levels of primary splenic fibroblasts infected with WT1 MCMV (MOI = 5) 8 hr post-infection. (**B**) IFNB1 mRNA levels of murine embryonic fibroblasts (MEF) from wildtype (WT) or indicated deficient mice were infected and analyzed as in (**A**). (**C**) IFNB1 mRNA levels in infected WT or cGAS-deficient primary splenic fibroblasts, analyzed as in (**A**). (**D**) Secreted IFNβ by WT or indicated gene deficient MEF, infected with MCMV (MOI = 0.5); supernatant was analyzed 48 hours p.i. by ELISA. Panels show representative experiments from two independent experiments performed in duplicate. WT, STING GT, and TBK1-, MAVS-,

*Figure 5 continued on next page*

*Figure 5 continued*

ZBP1-, DNA-PK-, and cGAS-deficient MEF represent data from two independent MEF preparations. Error bars indicate SEM; ns, not significant, *p<0.05, **p<0.01, ***p<0.001.
The online version of this article includes the following source data for figure 5:

**Source data 1.** MCMV-induced fibroblast IFNβ is triggered by cGAS-STING-dependent but MyD88-Trif-MAVS-independent mechanisms.

target cell rejection increased 3 days post-MCMV infection from 50% to 80% (*Figure 6C*). MHC-I-deficient cell ('missing self') rejection was higher and increased from 30% to 90%. MyD88 KO or STING GT mice did not display substantial differences in target cell rejection, but DKO mice substantially decreased NK cell cytolytic capacity with m157-specific rejection showing levels of uninfected mice. Similarly, MHC-I-deficient rejection was decreased in double versus single deficient mice. Together, these data indicate that both MyD88 and STING-dependent pathways contribute to NK cell cytolytic potential, albeit that MyD88 predominantly affects production of Prf and GzB.

## Discussion

Herein we describe that MCMV infection can be sensed by both STING and MyD88-dependent pathways which contribute to viral control in response to lethal challenge. While we confirmed the strong role of MyD88 in the hematological compartment, especially in splenic pDCs, we found that STING contributes in both the hematological and the previously unappreciated stromal cell compartment. Using primary splenic stromal cell cultures, we identified a role for cGAS-STING-dependent, but MyD88-independent IFN-I production in response to MCMV infection. Finally, we found that both MyD88 and STING-dependent pathways contribute to increased NK cell cytolytic function during infection. Thus, our findings indicate that cytomegalovirus infection is sensed by distinct sensing pathway depending on the infected cell type and that these pathways constitute a multi-layer antiviral defense.

Cytomegaloviruses have a broad tropism and a broad range of infected cell types have the capacity to produce IFN-I in response to infection. However, IFN-I production has been most well characterized in myeloid cells, including pDCs and Kupffer cells. IFN-I production by pDCs upon MCMV infection in vitro and in vivo is dependent on TLR9 and MyD88 (*Krug et al., 2004*; *Tomasello et al., 2018*; *Zucchini et al., 2008b*). Using IFNβ reporter mice, we were able to confirm that pDCs were the major source of IFNβ in the spleen and that this was dependent on MyD88. Furthermore, we observed that this IFN-I production was independent of STING. Early after infection, Kupffer cells in the liver produce IFN-I in a STING-dependent, but TLR-independent manner (*Tegtmeyer et al., 2019*). Hepatocytes are a major target for infection by MCMV (*Sacher et al., 2008*), yet they do not induce detectable levels of IFNβ (*Tegtmeyer et al., 2019*). However, hepatocytes do not express STING (*Thomsen et al., 2016*), likely explaining the lack of IFNβ production in hepatocytes in response to MCMV infection. In our bone marrow chimeras, we observed a role for STING in the radioresistant compartment in the liver. Hepatic stromal cells, including endothelial cells, are infected with MCMV (*Sacher et al., 2008*), providing likely contributors, apart from hepatocytes per se, to control MCMV in a STING-dependent manner in the liver.

MCMV is known to induce strong MyD88-dependent IFN-I responses in pDCs (*Krug et al., 2004*; *Zucchini et al., 2008a*). Consistent with these results, we observed both *Ifna* and *Ifnb1* transcripts, particularly in infected CD45$^+$CD11c$^+$ cells. We also observed equal expression of *Ifna* in the uninfected compartment. IFNα as well as IL-12 have been shown previously to be primarily produced by purified pDCs that were not productively infected (*Dalod et al., 2003*). Intriguingly, *Ifnb1* transcripts were much lower in the uninfected compartment as compared to the infected compartment. There was a trend in increased *Ifnb1* transcripts in uninfected CD11c$^+$ cells from infected versus uninfected mice, but this did not reach statistical significance. The observed difference in IFNα versus IFNβ production is remarkable; specifically, the IFN-I production by the pDCs that are not productively infected could be due to multiple reasons. IFN-I can act to produce more IFN-I as part of a positive feedback loop (*McNab et al., 2015*). However, this loop does not appear to play a major role in pDC after MCMV infection, as IFN-I receptor-deficient pDC in mixed bone marrow chimeras express similar levels of *Ifna* and *Ifnb1* as WT pDC (*Tomasello et al., 2018*). The *Ifna* production by the

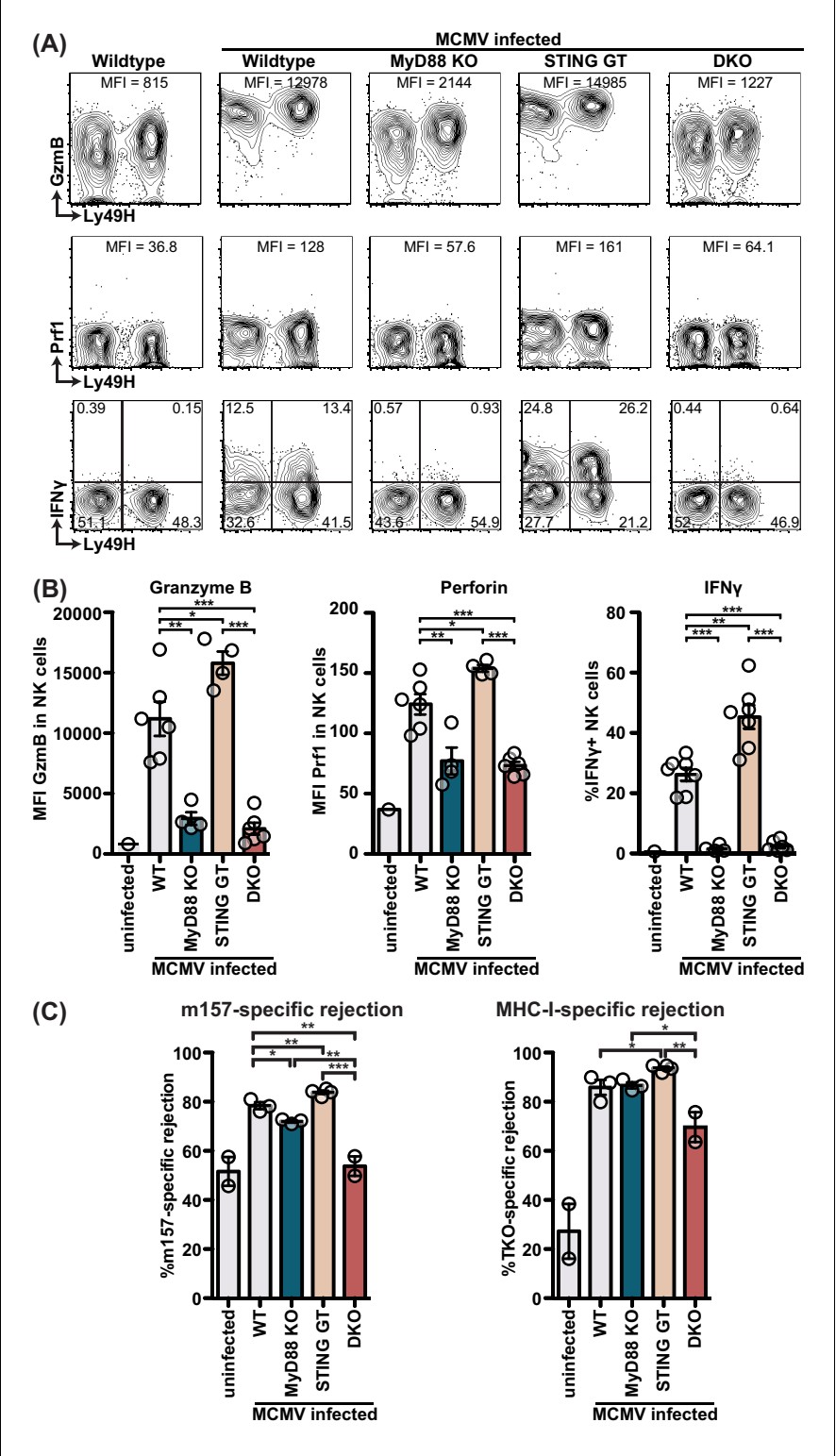

**Figure 6.** MyD88 and STING are required for NK cell cytolytic capacity during MCMV infection. (**A and B**) Mice deficient in MyD88 and/or STING were infected with MCMV and 2 days later splenocytes were harvested and analyzed for GzmB, Prf1, and IFNγ expression by FACS. Representative contour plots of individual mice are shown in (**A**) and quantification for multiple mice is shown in (**B**). (**C**) Differentially labelled WT, m157-Tg and MHC-I deficient splenocytes were adoptively transferred into indicated day 3-infected mice. Specific rejection was

*Figure 6 continued on next page*

*Figure 6 continued*

analyzed 3 hr post-transfer in the spleen. Representative experiments from two independent experiments per panel are shown. Error bars indicate SEM; ns, not significant, *p<0.05, **p<0.01, ***p<0.001.

The online version of this article includes the following source data for figure 6:

**Source data 1.** MyD88 and STING are required for NK cell cytolytic capacity during MCMV infection.

uninfected CD11c$^+$ cells could be the result of non-productive infection that did not result in IE1expression that was used to detect infected GFP$^+$ cells. Alternatively, pDC may take up apoptotic bodies from infected cells or pDC may receive other signals delivered by neighboring infected cells. Which of these or other causes underlie IFN-I production by uninfected (GFP$^-$) cells warrants further investigation.

We observed a stronger effect of STING in the splenic stromal cell compartment compared to the liver stromal cell compartment. IFN-I produced by splenic stromal cells have previously been reported to be dependent on lymphotoxin (LT) β expression by B cells, independent of TLR signaling (*Sacher et al., 2008*). Consistent with these findings, our findings revealed that stromal cell IFN-I is cGAS-STING-dependent. Besides the anti-viral role for STING in the stromal cell compartment, we observed that IFN-I production by infected primary splenic stromal cells was cGAS-STING-dependent. The primary stromal cells did not require interactions with B cells to produce IFN-I in vitro. However, it remains to be determined how LT intersects with the STING pathway in vivo, particularly since LT has been reported to be required for cell survival during MCMV infection (*Banks et al., 2005*), potentially providing a window where infected stromal cells survive long enough to produce IFN-I. Additional studies are required to further define these experimentally complex interactions.

Herpesviruses dedicate a large part of their genome to immune evasion strategies, including strategies that act on cellular immunity and intrinsic cellular defenses (*Powers et al., 2008*). MCMV has been reported to interfere with the DNA sensing pathway at different steps, m152 binds to STING and interferes with its trafficking (*Stempel et al., 2019*), whereas m35 targets NFκB-mediated transcription (*Chan et al., 2017*). Deletion of these ORFs individually in MCMV resulted in stronger IFN-I responses upon infection in vivo. Infection with MCMV deleted in both ORFs potentially increased STING-dependent viral control and may facilitate visualization of the IFN-I production by the cell types under study. Despite these immune evasion strategies, WT MCMV induced an IFN-I response that was potent enough to control infection, therefore we chose to use WT MCMV in the current study.

Infection with lethal dose of MCMV resulted in about a third of the MyD88-deficient mice succumbing to infection, which is consistent with previously published results (*Delale et al., 2005*). The impact of MyD88-deficiency on mortality and viral load was greater than STING-deficiency, which coincided with a bigger proportion of the IFN-I response was dependent on MyD88 as compared to STING. Consistent with these findings, markers for NK cell activation were mostly MyD88-dependent, whereas STING did not have a strong effect. Taken together, our data suggest that MyD88 induces a strong IFN-I response, whereas STING mediates a more moderate IFN-I response, likely contributing to more moderate morbidity in its absence as compared to MyD88-deficency. A recent study did not observe a lethality phenotype in mice deficient in MyD88 and TRIF, unless mice also lacked MAVS (*Tegtmeyer et al., 2019*). The latter study utilized tissue culture-derived MCMV in contrast to salivary gland extracted MCMV in the former and our study. Additionally, Tegtmeyer et al. used a mutant MCMV that lacked m157, whereas we used m157-sufficient virus for infections monitoring survival. Since IFN-I impacts NK cell-dependent MCMV-control via m157 recognition (*Parikh et al., 2015*), the use of WT MCMV allowed us to evaluate the effect of the virus-sensing pathways on NK cell function.

MyD88 KO mice have been reported to have a delay in initial weight loss (*Delale et al., 2005*), we observed a similar delay in weight loss in MyD88 KO and in DKO mice. STING did not cause any such overt immunopathology, indicating that this phenomenon may be specific to MyD88-dependent pathways. Moreover, the immunopathology is likely independent of IFN-I as a recent study showed that IFNAR-deficient mice still displayed early weight loss (*Tegtmeyer et al., 2019*). Additional research is required to better understand the MyD88-mediated initial weight loss in MCMV infected animals.

IFN-I and IL-12 produced in response to MCMV infection are required for full NK cell cytolytic capacity, through induction of GzB and Prf (*Parikh et al., 2015*). Consistent with previous reports (*Krug et al., 2004*; *Puttur et al., 2016*), we found that MyD88-deficient mice expressed low levels of NK cell GzB, Prf and IFNγ in response to MCMV infection but NK cytolytic potential in vivo was not substantially affected, consistent with previously published results (*Cocita et al., 2015*). However, MCMV-infected mice deficient in both STING and MyD88 displayed reduced NK cell cytolytic activity against m157-expressing and MHC-I-deficient target splenocytes. Thus, MyD88- and STING-dependent sensing of MCMV both contribute to signal to NK cells to enhance their cytolytic function in order to efficiently clear MCMV-infected target cells.

## Materials and methods

**Key resources table**

| Reagent type (species) or resource | Designation | Source or reference | Identifiers | Additional information |
|---|---|---|---|---|
| Strain, C57BL/6 background (*Mus musculus*) | C57BL/6 | Charles River Laboratories | 556; RRID:MGI:2160593 | |
| Strain, BALB/c background (*Mus musculus*) | BALB/c | Charles River Laboratories | 555; RRID:MGI:2160915 | |
| Strain, C57BL/6 background (*Mus musculus*) | STING golden ticket | Jackson Laboratories | 017537; RRID:IMSR_JAX:017537 | |
| Strain, C57BL/6 background (*Mus musculus*) | IFNβ-YFP reporter mice | Jackson Laboratories | 010818; RRID:IMSR_JAX:010818 | |
| Strain, C57BL/6 background (*Mus musculus*) | DNA-PK SCID | Jackson Laboratories | 001913; RRID:IMSR_JAX:001913 | |
| Strain, C57BL/6 background (*Mus musculus*) | B2m KO | Jackson Laboratories | 002087; RRID:IMSR_JAX:002087 | |
| Strain, C57BL/6 background (*Mus musculus*) | M157-Tg | *Tripathy et al., 2008* | | |
| Strain, C57BL/6 background (*Mus musculus*) | H-2K$^b$ x H-2D$^b$ KO | Taconic | 4215; RRID:IMSR_TAC:4215 | |
| Strain, C57BL/6 background (*Mus musculus*) | MyD88 KO | S. Akira | RRID:MGI:3577712 | through the JCRB Laboratory Animal Resource Bank of the National Institute of Biomedical Innovation |
| Strain, C57BL/6 background (*Mus musculus*) | TBK1 KO | S. Akira | nbio156; RRID:MGI:3053427 | through the JCRB Laboratory Animal Resource Bank of the National Institute of Biomedical Innovation |
| Strain, C57BL/6 background (*Mus musculus*) | ZBP1 KO | S. Akira | nbio155; RRID:MGI:3776852 | through the JCRB Laboratory Animal Resource Bank of the National Institute of Biomedical Innovation |
| Strain, C57BL/6 background (*Mus musculus*) | IPS1 KO | Michael Gale | | |

*Continued on next page*

*Continued*

| Reagent type (species) or resource | Designation | Source or reference | Identifiers | Additional information |
|---|---|---|---|---|
| Strain, C57BL/6 background (*Mus musculus*) | cGAS KO | Herbert Virgin | | |
| Other | IRF3/7 KO MEF | Michael Diamond | | Primary murine embryonic fibroblasts. |
| Other | STING KO MEF | Glen Barber | | Primary murine embryonic fibroblasts. |
| Other | MyD88xTRIF KO MEF | This paper | | Primary murine embryonic fibroblasts. See Materials and methods, Section 2 |
| Other | TBK1 KO MEF | This paper | | Primary murine embryonic fibroblasts. See Materials and methods, Section 2 |
| Other | TBK1 HET MEF | This paper | | Primary murine embryonic fibroblasts. See Materials and methods, Section 2 |
| Other | STING GT MEF | This paper | | Primary murine embryonic fibroblasts. See Materials and methods, Section 2 |
| Other | MAVS (IPS1) KO MEF | This paper | | Primary murine embryonic fibroblasts. See Materials and methods, Section 2 |
| Other | MAVS (IPS1) KO MEF | This paper | | Primary murine embryonic fibroblasts. See Materials and methods, Section 2 |
| Other | MAVS (IPS1) KO MEF | This paper | | Primary murine embryonic fibroblasts. See Materials and methods, Section 2 |
| Other | ZBP1 KO MEF | This paper | | Primary murine embryonic fibroblasts. See Materials and methods, Section 2 |
| Other | DNA-PK$^{SCID}$ MEF | This paper | | Primary murine embryonic fibroblasts. See Materials and methods, Section 2 |
| Other | cGAS KO MEF | This paper | | Primary murine embryonic fibroblasts. See Materials and methods, Section 2 |
| Virus (*murine cytomegalovirus*) | MCMV WT1 | *Cheng et al., 2010* | | |
| Virus (*murine cytomegalovirus*) | MCMV GFP | *Henry et al., 2000* | | |

*Continued on next page*

Continued

| Reagent type (species) or resource | Designation | Source or reference | Identifiers | Additional information |
|---|---|---|---|---|
| Sequence-based reagent | MCMV IE1 | IDT DNA | TAQman assay | Forward: 5'-CCCTCTCCTAACTCTCCCTTT-3'; Reverse: 5'-TGGTGCTCTTTTCCCGTG −3'; Probe: 5'-TCTCTTGCCCCGTCCTGAAAACC-3' |
| Sequence-based reagent | ACTB | IDT DNA | TAQman assay | Forward: 5'-AGCTCATTGTAGAAGGTGTGG-3'; Reverse: 5'-GGTGGGAATGGGTCAGAAG-3'; Probe: 5'-TTCAGGGTCAGGATA CCTCTCTTGCT-3' |
| Sequence-based reagent | IFNB1 | Thermo Fisher Scientific | TAQman assay | Mm00439546_s1 |
| Sequence-based reagent | (pan)Ifna | IDT DNA | TAQman assay | Forward: 5'-CTTCCACAGGATC ACTGTGTACCT-3'; Reverse: 5'-TTCTGCTC TGACCACCTCCC-3'; Probe: 5'-AGAGAGAAGAAACACAGCCC CTGTGCC-3' |
| Sequence-based reagent | GAPDH | Thermo Fisher Scientific | TAQman assay | Mm99999915_g1 |
| Antibody | Anti-mouse NK1.1 PE-Cy7 (Mouse monoclonal) | Thermo Fisher Scientific | Cat#: 25-5941-82; RRID:AB_469665 | FACS (1:100) |
| Antibody | Anti-mouse NKp46 PerCP-eFluor710 (Rat monoclonal) | Thermo Fisher Scientific | Cat#: 46-3351-82; RRID:AB_1834441 | FACS (1:100) |
| Antibody | Anti-mouse CD3 APC-eFluor780 (Armenian hamster monoclonal) | Thermo Fisher Scientific | cat# 47-0031-82, RRID:AB_11149861 | FACS (1:100) |
| Antibody | Anti-mouse CD19 APC-eFluor780 (Rat monoclonal) | Thermo Fisher Scientific | Cat# 47-0193-82, RRID:AB_10853189 | FACS (1:100) |
| Antibody | Ly49H FITC (Mouse monoclonal) | Made in-house | | FACS (1:200) |
| Antibody | Anti-mouse CD31 PE (Rat monoclonal) | Thermo Fisher Scientific | Cat# 12-0311-83, RRID:AB_465633 | FACS (1:100) |
| Antibody | Anti-mouse PDCA1 PE (Mouse monoclonal) | Thermo Fisher Scientific | Cat# 12-3171-81, RRID:AB_763427 | FACS (1:50) |
| Antibody | Anti-mouse gp38 PE-Cy7 (Syrian hamster monoclonal) | Thermo Fisher Scientific | Cat# 25-5381-82, RRID:AB_2573460) | FACS (1:100) |
| Antibody | Anti-mouse CD45 APC (Rat monoclonal) | Thermo Fisher Scientific | Cat# 17-0451-83, RRID:AB_469393) | FACS (1:50) |
| Antibody | Anti-mouse CD11c APC-eFluor780 (Armenian hamster monoclonal) | Thermo Fisher Scientific | Cat# 47-0114-82, RRID:AB_1548652) | FACS (1:50) |
| Antibody | Anti-mouse Ly49H APC (Mouse monoclonal) | Thermo Fisher Scientific | Cat# 17-5886-82, RRID:AB_10598809 | FACS (1:100) |
| Antibody | Anti-mouse Perforin PE (Rat monoclonal) | Thermo Fisher Scientific | Cat# 12-9392-82, RRID:AB_466243 | FACS (1:50) |

*Continued*

| Reagent type (species) or resource | Designation | Source or reference | Identifiers | Additional information |
|---|---|---|---|---|
| Antibody | Anti-mouse Granzyme B APC (Mouse monoclonal) | Thermo Fisher Scientific | Cat# MHGB05, RRID:AB_10373420 | FACS (1:100) |
| Antibody | Anti-mouse IFNg eFluor450 (Rat monoclonal) | Thermo Fisher Scientific | Cat# 48-7311-82, RRID:AB_1834366 | FACS (1:100) |
| Commercial assay or kit | Mouse IFNB ELISA | Biolegend | 439407 | |
| Commercial assay or kit | Cytofix/ Cytoperm kit | BD Biosciences | 554714 | |
| Software, algorithm | Prism | Graphpad | RRID: SCR_002798 | |
| Software, algorithm | Flowjo | Treestar Inc | RRID:SCR_008520 | |
| Other | Viability stain eFluor 506 | Thermofisher Scientific | 65-0866-14 | FACS (1:1000) |

## Mice

C57BL/6 (stock number 556) and BALB/c (555) mice were purchased from Charles River Laboratories. The following mouse strains were purchased from Jackson Laboratories: STING golden ticket (*Sting1 <gt >* ; 017537), IFNβ-YFP reporter mice (*Ifnb1*; 010818), DNA-PK SCID (*Prkdc <scid >* ; 001913), and β2m KO (*B2m*; 002087) all on the C57BL/6 background. m157-Tg mice were generated and maintained in-house on the C57BL/6 background (*Tripathy et al., 2008*). H-2K$^b$ KO x H-2D$^b$ KO (*H2-k1 x H2-d1*; 4215) mice on the C57BL/6 background were purchased from Taconic Farms. MyD88 KO (*Myd88*), TBK1 KO (*Tbk1*; nbio156), and ZBP1 KO (*Zbp1*; nbio155) mice were kindly provided by S. Akira (Osaka University, Osaka, Japan) through the JCRB Laboratory Animal Resource Bank of the National Institute of Biomedical Innovation (*Adachi et al., 1998*; *Hemmi et al., 2004*; *Ishii et al., 2008*) and were maintained on a C57BL/6 background. IPS1 KO (*Mavs*) mice on the C57BL/6 background were kindly provided by Michael Gale (University of Washington, Seattle, WA, USA). Mice deficient for cGAS (*Cgas*) were kindly provided by Herbert Virgin (Vir Biotechnology, San Francisco, CA, USA) (*Schoggins et al., 2014*). Triple MHC Class I KO mice (TKO) were generated by crossing β2m KO mice to H-2K$^b$ KO x H-2D$^b$ KO mice. STING GT mice were crossed to MyD88 KO to generate DKO mice. Subsequently DKO and single KO mice were crossed with IFNβ-YFP reporter to generate IFNβ-YFP on the various KO backgrounds. All mice were maintained in-house in accordance with institutional ethical guidelines. Age- and sex-matched mice were used in all experiments.

## Cell lines

3T12 cells (ATCC CCL-164) were maintained in DMEM supplemented with newborn calf serum, L-glutamine, penicillin, and streptomycin and were used for production of tissue culture derived MCMV and tittering of virus stocks. All MEF were maintained in RPMI supplemented with fetal bovine serum, L-glutamine, penicillin, and streptomycin. IRF3/7 KO MEF were kindly provided by Michael S Diamond (Washington University in St Louis, MO, USA). STING KO MEF have been described before (*Ishikawa et al., 2009*). All other MEF lines were generated from day 11.5–13.5 embryos, at least two independent lines were generated per genotype. To generate splenic fibroblasts, spleens were minced and digested with Liberase TL, adherent cells were cultured for 3–6 weeks to obtain pure fibroblast populations.

## In vivo virus infections

For in vivo studies salivary gland MCMV (sg-MCMV) of the WT-1 strain, a subcloned Smith strain (*Cheng et al., 2010*), was used for infections unless otherwise indicated. Where indicated, MCMV that expressed GFP under the IE1 promotor was used to visualize infected cells (*Henry et al., 2000*). This reporter virus contained a mutation in m157. All viral strains for in vivo infections were

propagated in BALB/c mice; virus was isolated from salivary glands and titers were determined as previously described (*Brune et al., 2001*; *Jonjic, 2001*). Mice were infected with indicated dose of MCMV intraperitoneally in 200 µl PBS. For survival studies weight was monitored daily and mice were sacrificed when more than 30% of initial weight was lost, in accordance to animal protocol. Viral load analysis was performed as previously described (*Parikh et al., 2015*). Briefly, RNA-free organ DNA was isolated using Puregene extraction kit (Qiagen). 160 ng DNA was quantified for MCMV IE1 (Forward: 5'-CCCTCTCCTAACTCTCCCTTT-3'; Reverse: 5'-TGGTGCTCTTTTCCCGTG −3'; Probe: 5'-TCTCTTGCCCCGTCCTGAAAACC-3'; IDT DNA) and host *Actb* (Forward: 5'- AGC TCATTGTAGAAGGTGTGG-3'; Reverse: 5'- GGTGGGAATGGGTCAGAAG-3'; Probe: 5'-TTCAGGG TCAGGATACCTCTCTTGCT-3'; IDT DNA) against plasmid standard curves using TAQman universal master mix II on a StepOnePlus real time PCR system (Thermo Fisher Scientific).

## Bone marrow chimeras
C57BL/6 and STING GT mice were irradiated with 950 rad by an x-ray irradiator and were intravenously with 5 million of the indicated genotype donor bone marrow cells. Chimeric mice were given antibiotic water (sulfamethoxazole/trimethoprim) for 4 weeks. 6 weeks post-irradiation mice were infected with MCMV and analyzed for viral load at 5 days p.i. We observed greater sensitivity of reconstituted BM chimeric mice to infections than mice not subjected to the BM transplant procedure in our facility so we infected reconstituted mice with a lower dose of MCMV (20,000 PFU) as compared to non-chimeric mice.

## In vitro virus infections
For in vitro studies, pelleted tissue culture-derived MCMV was prepared and viral titers were determined as previously described (*Brune et al., 2001*). 200,000 cells were plated in a 6-well plate overnight and were infected with 200 µl of MCMV at MOI five for RNA analysis and MOI 0.5 for supernatant analysis for 1 hr, after which wells were washed with PBS to remove free virus and 2 ml fresh culture media was added. Cells were lysed in the wells with 1 ml trizol after an additional 5 hr culture for RNA analysis. Samples were stored at −80℃ until analysis. Supernatants were harvested 48 hr after culture and analyzed for IFNβ by ELISA (Biolegend) according to manufacturer protocol.

## Flow cytometry
Fluorescent-labeled antibodies used were anti-NK1.1 (clone PK136), anti-NKp46 (29A1.4), anti-CD3 (145–2 C11), anti-CD19 (eBio1D3), anti-CD31 (390), anti-PDCA1 (eBio129c), anti-gp38 (eBio8.1.1), anti-CD45 (30-F11), anti CD11c (N418), anti-Ly49H (3D10), anti-Perforin (eBioOMAK-D), anti-Granzyme B (GB12), and anti-IFNγ (XMG1.2), all from Thermo Fisher Scientific. For analysis of splenic dendritic and stromal cells, spleens were digested with 1 mg/ml Liberase TL and DNAse -I (Millipore Sigma) for 45 min with mechanical dissociation with a pipette every 15 min to obtain a single cell suspension. For analysis of NK cells, spleens were crushed through a 70 µm cell strainer to obtain a single cell suspension. Red blood cells (RBC) in all samples were lysed with RBC lysis buffer. Cells for analysis were first stained with fixable viability day (Thermo Fisher Scientific). Subsequently, cell surface molecules were stained in 2.4G2 hybridoma supernatant to block Fc receptors. For intracellular staining, cells were fixed and stained intracellularly using the Cytofix/Cytoperm kit (BD Biosciences) according to manufacturer's instructions. Samples were acquired using FACSCanto (BD Biosciences) and analyzed using FlowJo software (Treestar). NK cells were defined as Viability-NK1.1$^+$CD3$^-$CD19$^-$. Where indicated, cells were sorted on a FACSaria (BD Biosciences) into media and subsequently lysed in Trizol for RNA analysis.

## RNA analysis
RNA was isolated from cultured or sorted cells using Trizol according to manufacturer instruction (Thermo Fisher Scientific). Contaminating DNA was removed using Turbo DNAse, and cDNA was synthesized using Superscript III using oligo(dT) (Thermo Fisher Scientific). Quantification was performed for *Ifnb1* (Mm00439546_s1; Thermo Fisher Scientific), (pan)*Ifna* (Forward: 5'-CTTCCA-CAGGATCACTGTGTACCT-3'; Reverse: 5'-TTCTGCTC tgaccacctccc-3'; Probe: 5'-AGAGAGAAGAAACACAGCCC CTGTGCC-3'; IDT DNA) (*Samuel and Diamond, 2005*) and *Gapdh*

(Mm99999915_g1; Thermo Fisher Scientific) against plasmid or pooled standard curves using TAQman universal master mix II on a StepOnePlus real time PCR system (Thermo Fisher Scientific).

## In vivo cytotoxicity assay

Target splenocytes were isolated from C57BL/6, m157-Tg, and MHC-I deficient (TKO) mice and differentially labelled with CFSE, CellTrace violet, and CellTrace far red (Thermo Fisher Scientific). Target cells were mixed at a 1:1:1 ratio and $3 \times 10^6$ target cells were injected i.v. into naïve or day 3 MCMV-infected mice. 3 hr after challenge splenocytes were harvested and stained. The ratio of target (m157-tg or TKO) to control (C57BL/6) viable $CD19^+$ cells was determined by flow cytometry. Target cell rejection was calculated using the formula [(1−(Ratio(target:control)$_{sample}$/Ratio(target:control)$_{NK\ depleted}$))×100]. Average of two NK1.1-depleted mice served as control.

## Statistical analysis

Statistical analysis was performed using Prism (GraphPad software). Survival curves were compared using Log-Rank (Mantel-Cox) tests, other comparisons were performed using one-way ANOVA with Bonferroni's multiple comparisons tests to calculate P values. Error bars in figures represent the SEM. Statistical significance was indicated as follows: ****, $p<0.0001$; ***, $p<0.001$; **, $p<0.01$; *, $p<0.05$; ns, not significant.

## Acknowledgements

We thank Beatrice Plougastel-Douglas for critically reading the manuscript.

This work was supported by NIH grant R01-AI131680 to WMY and SJP was supported by the Netherlands Organisation for Scientific Research (Rubicon grant 825.11.004).

## Additional information

### Funding

| Funder | Grant reference number | Author |
| --- | --- | --- |
| National Institute of Allergy and Infectious Diseases | R01-AI131680 | Wayne M Yokoyama |
| Nederlandse Organisatie voor Wetenschappelijk Onderzoek | Rubicon grant 825.11.004 | Sytse J Piersma |

The funders had no role in study design, data collection and interpretation, or the decision to submit the work for publication.

### Author contributions

Sytse J Piersma, Conceptualization, Funding acquisition, Investigation, Visualization, Methodology, Writing - original draft, Writing - review and editing; Jennifer Poursine-Laurent, Liping Yang, Investigation, Methodology, Writing - review and editing; Glen N Barber, Resources, Writing - review and editing; Bijal A Parikh, Investigation, Writing - review and editing; Wayne M Yokoyama, Conceptualization, Formal analysis, Supervision, Writing - original draft, Writing - review and editing

### Author ORCIDs

Sytse J Piersma https://orcid.org/0000-0002-5379-3556
Wayne M Yokoyama http://orcid.org/0000-0002-0566-7264

### Ethics

Animal experimentation: This study was performed in strict accordance with the recommendations in the Guide for the Care and Use of Laboratory Animals of the National Institutes of Health. All of the animals were handled according to the approved institutional animal care and use committee (IACUC) protocol (#20180293). The protocol was approved by the Animal Studies Committee of Washington University.

Decision letter and Author response

Decision letter https://doi.org/10.7554/eLife.56882.sa1

Author response https://doi.org/10.7554/eLife.56882.sa2

# Additional files

## Supplementary files
- Transparent reporting form

## Data availability
All data generated or analysed during this study are included in the manuscript and supporting files. Source data files have been provided for all figures.

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
