## [Decision Letter]

**Acceptance summary:**

This paper describes the role of TLR-MyD88 and cGAS-STING pathways in control of cytomegalovirus infection in different cells/tissues. Using the model of bone marrow chimeras, the authors demonstrated the capacity of both pathways in virus control in hematopoietic cells, while virus control in splenic stromal cells was dependent on cGAS-STING pathway, but independent of TLR-MyD88 pathway. Also, CMV-induced NK cell cytotoxicity in vivo depends on both the cGAS-STING and the TLR-MyD88 pathway. While the role of MyD88-TLR pathway in hematopoetic cells was previously shown, the novelty of present study lies in the fact that MyD88 and STING contribute to virus control in distinct cell types that initiate downstream immune responses.

**Decision letter after peer review:**

Thank you for submitting your work entitled "Virus infection is controlled by cell type-specific sensing of murine cytomegalovirus through MyD88 and STING" for consideration by *eLife*. Your article has been evaluated by three reviewers, and the evaluation has been overseen by a Reviewing Editor and a Senior Editor. Our decision has been reached after consultation between the reviewers and editors.

Based on these discussions and the summary of reviews below, we reached the conclusion that your study is of general interest for *eLife*. However, although all reviewers are essentially positive, they raised several queries which require your attention and additional information before the manuscript can be considered for publication in *eLife*. The major concern of the reviewers is whether your study has sufficient novelty compared to published work to warrant publication in *eLife*. Therefore, please address all the queries raised by reviewers. Below please find the summary from reviewer's and editor's comments.

Previous studies have already shown cell type-dependent differences in the necessity of distinct pattern recognition receptors to sense MCMV infection, whereas this study specifically focuses on the role of STING- and MyD88-mediated responses in hematological and non-hematological cells. Using STING-deficient goldenticket, MyD88-/- and double knockout mice, the authors have shown that both pathways are critical to elicit an immune response strong enough to control MCMV infection. Moreover, in accordance with previous publications, the authors present data that stromal cells induce cGAS-STING-dependent responses to MCMV infection, while plasmacytoid dendritic cells (pDCs) rely on the presence of the adaptor protein MyD88 to combat infection. Furthermore, by analyzing NK activity markers, such as Perforin and Granzyme B, the authors showed that NK cell cytolytic activity is dependent on both, STING- and MyD88-mediated responses. Altogether, the main findings can be summarized as follows: (i) BM chimeras show dependence on two distinct pathways in hematopoietic cells triggered by endosomal and cytosolic recognition, mediated by the TLR-MyD88 and cGAS-STING pathways; (ii) stromal cells (mainly splenic) contribute to the cGAS-STING IFN-I pathway, but not TLR-MyD88 pathway; (iii) hematopoietic dendritic cells (mainly pDC) contribute to the TLR-MyD88-dependent IFN-I pathway, but not cGAS-STING pathway; (iv) NK cell-mediated cytotoxicity of m157+ and MHC-I- targets in MCMV-infected mice was dependent upon both the cGAS-STING and TLR-MyD88 pathways to facilitate enhanced clearance in vivo. In addition, they show that cell-autonomous production of IFNb can be distinguished from paracrine effects related to secondary IFNa production in hematopoietic cells, such as pDC, in the fact that the former is TLR-MyD88-dependent but cGAS-STING-independent.

The manuscript is well written and the experiments performed are solid. However, the reviewers seriously question the novelty as compared to current knowledge in the field (Schneider et al., Krug et al., Lio et al., Tegtmeyer et al.). Although the results on cytosolic sensing can be considered as novel, the results on MyD88-dependent sensing by plasmacytoid dendritic cells (pDC) are more confirmatory and therefore the authors should pay attention to properly cite important previous studies as well as accordingly interpret their own data.

Essential revisions:

1) Figure 1A/B: In Figure 1A, the authors display weight loss over 10 days post MCMV challenge, however, 70% of the DKO mice die after 6 days (Figure 1B). The figure legend states that 14 DKO mice were used for this figure. Of course, these are different data sets, however, mortality rate will be comparable. Are these data points shown in Figure 1A combined from 14 individual mice monitored over 10 days, or were 14 mice used in the beginning of the experiment and the later days only display the weight loss of the surviving mice?

2) Figure 2B and Figure 2—figure supplement 1A: The increased replication of MCMV in MyD88-deficient mice 5 dpi is not consistent. While Figure 2B shows a clear and significant increase in MCMV IE1 copy numbers in MyD88-/- mice, this difference is not apparent in Figure 2—figure supplement 1A. Can the authors clarify these differences, are there any other experimental changes (besides the viral dose which should not affect the results this much) which can cause these varieties.

3) Figure 3C: The IFNa transcription in uninfected GFP-DCs is very interesting. While the authors have concluded a potential feedforward loop, it would be straightforward to test if this response is induced due to the initial burst of type I IFN (IFNb) in other cell types. IFNAR KO mice would indicate if DCs are initiating IFNa transcription depending on type I IFN or if other cytokines are involved in this induction. The induction of IFNa in uninfected cells is underappreciated and suggests an important role to combat MCMV infection. This should be investigated further.

4) The vast majority of the pDC producing IFN-I and IL-12 during MCMV infection are not infected. The authors state in several instances that the pDC sensing MCMV in vivo to produce IFN-I are infected but the paper does not contain any clear and robust data supporting this statement. "we investigated anti-viral effects.… in distinct cell types that are infected.…", "findings indicate that cytomegalovirus infection is sensed by distinct sensing pathway depending on the infected cell type.…". The data used to document pDC infection is largely insufficient. In Figure 3A, pDC are defined as CD45+ CD11c(low) cells. This is not proper. Many other cell types share the same phenotype in the spleen, after MCMV infection, encompassing a sizeable fraction of classical monocytes, some macrophages, NK cells and mature classical DCs that have downregulated CD11c. Even if a fraction of pDC were infected, no data is shown to prove that these are the one producing IFN-I. The vast majority of the pDC sensing MCMV in vivo during the infection to produce IFN-I and IL-12 were previously shown not to be infected (Dalod et al., 2003. Hence, the authors must correct their wrong statements and cite this paper. This comment could be dealt by properly editing text in manuscript.

5) Uninfected pDC produce high levels of IFN-β. The authors state that "*Ifnb1* transcripts are specifically produced in response to viral detection in infected DCs as well as stromal cells". This is a misinterpretation of the data shown on Figure 3B. First, expression of the *Ifnb1* transcripts in GFP- CD45+CD11c+ cells is at least as high as in the CD45-GFP+ cells. Second, only a fraction of the GFP- CD45+CD11c+ cells are IFN-I-producing pDC, leading to an large underestimation of their *Ifnb1* expression. *Ifnb1* transcripts are highly expressed in pDC properly purified from the spleen of MCMV-infected mice (Zucchini et al., 2008a). Moreover, later in the paper, by using *Ifnb1*-EYFP reporter mice, the authors do show that "Approximately 20% of the pDCs were YFP+, indicating at least this percentage of infected pDCs produced IFNβ in response to MCMV infection, whereas much fewer cDCs produced IFNb because less than 1% of cDCs were YFP+ (Figure 4B)". Hence this observation from the authors combined with the known fact that IFN-I-producing pDC are not infected shows that uninfected pDC are the major expressers of *Ifnb1*. This comment could be dealt by properly editing text in manuscript.

6) Uninfected pDC do not require a feedforward loop to produce high levels of IFN-I.

The authors state "These data suggest that *Ifna* transcripts in uninfected DCs are produced (in part) as a feedforward loop (McNab et al., 2015)". This is incorrect. The vast majority of *Ifna* transcripts produced around 36 hours during MCMV infection arise from uninfected pDC that have been rigorously proven not to require the IFNAR1-dependent feedforward loop for this function (Tomasello et al., 2018). This comment could be dealt by properly editing text in manuscript.

---

## [Author Response]

[…] The manuscript is well written and the experiments performed are solid. However, the reviewers seriously question the novelty as compared to current knowledge in the field (Schneider et al., Krug et al., Lio et al., Tegtmeyer et al). Although the results on cytosolic sensing can be considered as novel, the results on MyD88-dependent sensing by plasmacytoid dendritic cells (pDC) are more confirmatory and therefore the authors should pay attention to properly cite important previous studies as well as accordingly interpret their own data.

We thank the editor and reviewers for their constructive comments. As pointed out by the editor, the novel findings of the manuscript included the following: contribution of both TLR-MyD88 and cGAS-STING pathway in the hematopoietic compartment on viral control using bone marrow chimeras; the role of mainly splenic stromal cells in viral control dependent on the cGAS-STING pathway, but not the TLR-MyD88 pathway; absence of the requirement of the cGAS-STING pathway for pDC IFN-I production; and MCMV-induced NK cell cytotoxicity in vivo depends on both the cGAS-STING and the TLR-MyD88 pathway.

We described the role of the MyD88-TLR pathway in pDC in this manuscript specifically to contrast with the cGAS-STING pathway, revealing its role in other cell types. As such, we agree that the findings on the role of the MyD88-TLR pathway in pDC are largely confirmatory. To better make our points, we de-emphasized the findings on the MyD88-TLR pathway in pDC by changing the title of the manuscript; indicating that the findings on pDC are confirmatory, for example in the Abstract; and including a new paragraph in the Discussion section that discusses additional references on pDC that were suggested by the reviewers.

As pointed out by the editor, our findings on the cGAS-STING pathway in stromal cells is novel. The role of the cGAS-STING pathway in control of MCMV infection has previously been reported by Lio et al. and by Tegtmeyer et al., but its role in different cellular compartments on viral control was not investigated in those studies. Schneider et al. previously investigated MCMV-induced IFN-I production by stromal cells. Our work now shows that stromal cell IFN-I is dependent on the cGAS-STING pathway and robustly contributes to viral control, in particular in the absence of MyD88 in the hematological compartment.

Taken together, we believe that our work provides new understanding of viral sensing, in particular in non-immune stromal cells, and its effects on viral control.

Our response to the specific comments from the reviewers are given below:

Essential revisions:1) Figure 1A/B: In Figure 1A, the authors display weight loss over 10 days post MCMV challenge, however, 70% of the DKO mice die after 6 days (Figure 1B). The figure legend states that 14 DKO mice were used for this figure. Of course, these are different data sets, however, mortality rate will be comparable. Are these data points shown in Figure 1A combined from 14 individual mice monitored over 10 days, or were 14 mice used in the beginning of the experiment and the later days only display the weight loss of the surviving mice?

The weight loss results are indeed from 14 individual mice at the start of the experiment and monitored over time. At later days (for MyD88 KO and DKO mice) displays only the weight loss of the surviving mice. These data can be accessed in the raw data file that was accompanied with the submission. To clarify, we added the following text to the legend of Figure 1: “The numbers indicate the number of mice at the start of the experiment, weight loss of surviving mice at each timepoint is plotted.”

2) Figure 2B and Figure 2—figure supplement 1A: The increased replication of MCMV in MyD88-deficient mice 5 dpi is not consistent. While Figure 2B shows a clear and significant increase in MCMV IE1 copy numbers in MyD88-/- mice, this difference is not apparent in Figure 2—figure supplement 1A. Can the authors clarify these differences, are there any other experimental changes (besides the viral dose which should not affect the results this much) which can cause these varieties.

We agree with the reviewer that this is an unexpected finding, but similar findings have been published before. Delale et al. reported that a 2.5-fold increase in MCMV dosing increased mortality in MyD88 KO and in TLR9 KO mice from 50% to 100% and from 20% to 100%, respectively (Delale et al., 2005). The experiments in Figure 2B and Figure 2—figure supplement 1 were independent experiments performed on different days. We did use the same batch of virus, the mice were all similar age, and both graphs are made up of 50% males and females, with no overt differences between sexes. We included these details in Author response table 1. We hypothesize that these findings are a result of the viral doses used in our studies, that are likely around the threshold to control infection.

The experimental details for these experiments were as follows:

Author response table 1

3) Figure 3C: The IFNa transcription in uninfected GFP-DCs is very interesting. While the authors have concluded a potential feedforward loop, it would be straightforward to test if this response is induced due to the initial burst of type I IFN (IFNb) in other cell types. IFNAR KO mice would indicate if DCs are initiating IFNa transcription depending on type I IFN or if other cytokines are involved in this induction. The induction of IFNa in uninfected cells is underappreciated and suggests an important role to combat MCMV infection. This should be investigated further.

We agree with the reviewer that the difference between *Ifna* and *Ifnb* in the GFP- DCs is a very interesting finding. Of note, as pointed out by the reviewer in comment 4, the sorted CD11c+ cells include more cell types than just DCs. In similar experiments, GFP- DCs have previously been reported to secrete IFNα and IL-12 (Dalod et al., 2003). Yet, unresolved is what could potentially underly differences between IFNa and IFNb. As pointed out by the reviewer in comment 6, the IFNa production has been reported to be independent of a feedforward loop (Tomasello et al., 2018). Several other causes may underly the IFNα production in GFP- DCs, including non-productive viral infection that did not result in IE1 expression, uptake of apoptotic bodies from infected cells, or other signals delivered by neighboring infected cells. These hypotheses warrant further investigation, but were beyond the scope of this manuscript. Also, please note that we have restrictions on laboratory work at our institution due to the COVID-19 pandemic.

We revised the Discussion to address these comments as well as comments 4-6 and added the following paragraph which also contains statements related to these other comments below.

“MCMV is known to induce strong MyD88-dependent IFN-I responses in pDCs (Krug et al., 2004; Zucchini et al., 2008a). […] Alternatively, pDC may take up apoptotic bodies from infected cells or pDC may receive other signals delivered by neighboring infected cells. Which of these or other causes underlie IFN-I production by uninfected (GFP^-^) cells warrants further investigation.”

4) The vast majority of the pDC producing IFN-I and IL-12 during MCMV infection are not infected. The authors state in several instances that the pDC sensing MCMV in vivo to produce IFN-I are infected but the paper does not contain any clear and robust data supporting this statement. "we investigated anti-viral effects.… in distinct cell types that are infected.…", "findings indicate that cytomegalovirus infection is sensed by distinct sensing pathway depending on the infected cell type.…". The data used to document pDC infection is largely insufficient. In Figure 3A, pDC are defined as CD45+ CD11c(low) cells. This is not proper. Many other cell types share the same phenotype in the spleen, after MCMV infection, encompassing a sizeable fraction of classical monocytes, some macrophages, NK cells and mature classical DCs that have downregulated CD11c. Even if a fraction of pDC were infected, no data is shown to prove that these are the one producing IFN-I.

We agree with the reviewer that the CD45+CD11c+ sorted population includes more cell types than just pDC. We changed DC designations to CD11c+ in the following paragraph of the Results section to avoid misinterpretations as pointed out by the reviewer:

“At 36-hours p.i. , the percentage of infected stromal cells increased substantially and infected CD45+CD11c+ cells were detected as well. […] Based on these data we chose to investigate the role of STING and MyD88 on IFNβ production by different cell types.”

The vast majority of the pDC sensing MCMV in vivo during the infection to produce IFN-I and IL-12 were previously shown not to be infected (Dalod et al., 2003). Hence, the authors must correct their wrong statements and cite this paper. This comment could be dealt by properly editing text in manuscript.

We commented on the IFN-I production by uninfected pDCs in the new paragraph mentioned in comment 3 above which includes sentences relevant to the production of IFN-I by GFP- CD45+CD11c+ cells.

“MCMV is known to induce strong MyD88-dependent IFN-I responses in pDCs (Krug et al., 2004; Zucchini et al., 2008a). […] Which of these or other causes underlie IFN-I production by uninfected (GFP^-^) cells warrants further investigation.”

5) Uninfected pDC produce high levels of IFN-β. The authors state that "Ifnb1 transcripts are specifically produced in response to viral detection in infected DCs as well as stromal cells". This is a misinterpretation of the data shown on Figure 3B. First, expression of the Ifnb1 transcripts in GFP- CD45+CD11c+ cells is at least as high as in the CD45-GFP+ cells. Second, only a fraction of the GFP- CD45+CD11c+ cells are IFN-I-producing pDC, leading to an large underestimation of their Ifnb1 expression. Ifnb1 transcripts are highly expressed in pDC properly purified from the spleen of MCMV-infected mice (Zucchini et al., 2008a). Moreover, later in the paper, by using Ifnb1-EYFP reporter mice, the authors do show that "Approximately 20% of the pDCs were YFP+, indicating at least this percentage of infected pDCs produced IFNβ in response to MCMV infection, whereas much fewer cDCs produced IFNb because less than 1% of cDCs were YFP+ (Figure 4B)". Hence this observation from the authors combined with the known fact that IFN-I-producing pDC are not infected shows that uninfected pDC are the major expressers of Ifnb1. This comment could be dealt by properly editing text in manuscript.

We agree with the reviewer that there seemed to be an increase in *Ifnb1* in GFP-CD45+CD11c+ cells in MCMV infected mice. However, the differences between GFP-CD45+CD11c+ cells form uninfected and infected mice were not significant, thus we were reluctant to interpret the *Ifnb1* expression in these populations. We commented on the *Ifnb1* expression by GFP- CD45+CD11c+ cells and the previously reported *Ifnb1* expression in pDC in a new paragraph mentioned in comment 3 above.

“MCMV is known to induce strong MyD88-dependent IFN-I responses in pDCs (Krug et al., 2004; Zucchini et al., 2008a). […] Which of these or other causes underlie IFN-I production by uninfected (GFP^-^) cells warrants further investigation.”

In the Results section we changed to the following paragraph:

“At 36-hours p.i., the percentage of infected stromal cells increased substantially and infected CD45+CD11c+ cells were detected as well. […] Based on these data we chose to investigate the role of STING and MyD88 on IFNβ production by different cell types.”

As mentioned by the reviewer we do not show that the IFNβ-reporter positive pDC were productively infected. To clarify this issue, we deleted this suggestion and changed the text in the Results section to:

“Approximately 20% of the pDCs were YFP^+^, indicating at least this percentage of pDCs produced IFNβ in response to MCMV infection, whereas much fewer cDCs produced IFNβ because less than 1% of cDCs were YFP^+^ (Figure 4B).”

6) Uninfected pDC do not require a feedforward loop to produce high levels of IFN-I.The authors state "These data suggest that Ifna transcripts in uninfected DCs are produced (in part) as a feedforward loop (McNab et al., 2015)". This is incorrect. The vast majority of Ifna transcripts produced around 36 hours during MCMV infection arise from uninfected pDC that have been rigorously proven not to require the IFNAR1-dependent feedforward loop for this function (Tomasello et al., 2018). This comment could be dealt by properly editing text in manuscript.

We thank the reviewer to bring this issue to our attention. We removed this sentence from the Results section. Instead we now discuss other possibilities that may underlie the production of IFN-I in GFP- cells. We hypothesize that several other causes may underly the IFNα production in GFP- DCs, including non-productive viral infection that did not result in IE1 expression, uptake of apoptotic bodies from infected cells, or other signals delivered by neighboring infected cells. These hypotheses warrant further investigation but are beyond the scope of the current manuscript.

We commented on the *Ifna* expression by GFP- DC in a new paragraph mentioned in comment 3 above.

“MCMV is known to induce strong MyD88-dependent IFN-I responses in pDCs (Krug et al., 2004; Zucchini et al., 2008a). […] Which of these or other causes underlie IFN-I production by uninfected (GFP^-^) cells warrants further investigation.”